# Distance-Based Regularisation of Deep Networks for Fine-Tuning

**Henry Gouk & Timothy M. Hospedales**
School of Informatics
University of Edinburgh
{henry.gouk,t.hospedales}@ed.ac.uk

**Massimiliano Pontil**
CSML, Istituto Italiano di Tecnologia
& Department of Computer Science, UCL
massimiliano.pontil@iit.it

## Abstract

We investigate approaches to regularisation during fine-tuning of deep neural networks. First we provide a neural network generalisation bound based on Rademacher complexity that uses the distance the weights have moved from their initial values. This bound has no direct dependence on the number of weights and compares favourably to other bounds when applied to convolutional networks. Our bound is highly relevant for fine-tuning, because providing a network with a good initialisation based on transfer learning means that learning can modify the weights less, and hence achieve tighter generalisation. Inspired by this, we develop a simple yet effective fine-tuning algorithm that constrains the hypothesis class to a small sphere centred on the initial pre-trained weights, thus obtaining provably better generalisation performance than conventional transfer learning. Empirical evaluation shows that our algorithm works well, corroborating our theoretical results. It outperforms both state of the art fine-tuning competitors, and penalty-based alternatives that we show do not directly constrain the radius of the search space.

## 1 Introduction

The ImageNet Large Scale Visual Recognition Challenges have resulted in a number of neural network architectures that obtain high accuracy when trained on large datasets of labelled examples (He et al., 2016; Tan and Le, 2019; Russakovsky et al., 2015). Although these models have been shown to achieve excellent performance in these benchmarks, in many real-world scenarios such volumes of data are not available and one must resort to fine-tuning an existing model: taking the weights from a model trained on a large dataset, to initialise the weights for a model that will be trained on a small dataset. The assumption being that the weights from the pre-trained model provide a better initialisation than randomly generated weights. Approaches for fine-tuning are typically ad hoc, requiring one to experiment with many problem-dependent tricks, and often a process that will work for one problem will not work for another. Transforming fine-tuning from an art into a well principled procedure is therefore an attractive prospect. This paper investigates, from both a theoretical and empirical point of view, the impact of different regularisation strategies when fine-tuning a pre-trained network for a new task.

Existing fine-tuning regularisers focus on augmenting the cross entropy loss with terms that indirectly or directly penalise the distance the fine-tuned weights move from the pre-trained values. The intuition behind this seems sensible—the closer the fine-tuned weights are to the pre-trained weights, the less information is forgotten about the source dataset—but it is not obvious how this idea should be translated into an effective algorithm. One should expect that the choice of distance metric is quite important, but existing methods exclusively make use of Euclidean distance (Li et al., 2019; 2018) without any theoretical or empirical justification regarding why that metric was chosen. These methods achieve only a small improvement in performance over standard fine-tuning, and it is reasonable to expect that using a metric more suited to the weight space of neural networks would lead to greater performance. Moreover, while the use of penalty terms to regularise neural networks is well established, the impact of using penalties vs constraints as regularisers has not been well studied in the context of deep learning.

In order to study the generalisation error of fine-tuned models, we derive new bounds on the empirical Rademacher complexity of neural networks based on the distance the trained weights move from their initial values. In contrast to existing theory (e.g., Neyshabur et al. (2018); Bartlett et al. (2017);

Long and Sedghi (2019)), we do not resort to covering numbers or make use of distributions over models to make these arguments. By deriving two bounds utilising different distance metrics, but proved with the same techniques, we are able to conduct a controlled theoretical comparison of which metric one should use as the basis for a fine-tuning regularisation scheme. Our findings show that a metric based on the maximum absolute row sum (MARS) matrix norm is a more suitable measure of distance in the parameter space of convolutional neural networks than Euclidean distance. Additionally, we challenge the notion that using a penalty term to encourage the fine-tuned weights to lie near the pre-trained values is the best way to restrict the effective hypothesis class. We demonstrate that the equivalence of penalty methods and constraint methods in the case of linear models (Oneto et al., 2016) does not translate to the context of deep learning. As a result, using projected stochastic subgradient methods to constrain the distance the weights in each layer can move from the initial settings can lead to improved performance.

Several regularisation methods are proposed, with the aim of both corroborating the theoretical analysis with empirical evidence, and improving the performance of fine-tuned networks. One of these approaches is a penalty-based method that regularises the distance from initialisation according to the MARS-based distance metric. The other two techniques make use of efficient projection functions to enforce constraints on the Euclidean and MARS distance between the pre-trained and fine-tuned weights throughout training. The experimental results demonstrate that projected subgradient methods improve performance over using penalty terms, and that the widely used Euclidean metric is typically not the best choice of metric to measure distances in network parameter space.

## 2 RELATED WORK

The idea of developing an algorithm to restrict the distance of weights from some unbiased set of reference weights has been explored in various forms to improve the performance of fine-tuned networks. Li et al. (2018) presented the $\ell^2$-SP regulariser, which consists of adding a term to the objective function that penalises the squared $\ell^2$ distance of the trained weights from the initial weights. This is based on an idea initially made use of when performing domain adaptation, where it was applied to linear support vector machines (Yang et al., 2007). The subsequent work of Li et al. (2019) follows the intuition that the *features* produced by the fine-tuned network should not differ too much from the pre-trained features. They also use Euclidean distance, but to measure distance between feature vectors rather than weights. The idea is extended to incorporate an attention mechanism that weights the importance of each channel. The method is implemented by adding a penalty term to the standard objective function. In contrast to these approaches, we solve a constrained optimisation problem rather than adding a penalty, and we demonstrate that the MARS norm is more effective than the Euclidean norm when measuring distance in weight space.

Many recent meta-learning algorithms also make use of idea that keeping fine-tuned weights close to their initial values is desirable. However, these approaches typically focus on developing methods for learning the initial weights, rather than working with pre-specified initial weights. The model-agnostic meta-learning approach (Finn et al., 2017) does this by simulating few-shot learning tasks during the meta-learning phase in order to find a good set of initial weights for a neural network. Once the learned algorithm is deployed, it adapts to new few-shot learning tasks by fine-tuning the initial weights for a small number of iterations. Denevi et al. (2018) proposes a modified penalty term for ridge regression where, instead of penalising the distance of the parameters from zero, they are regularised towards a bias vector. This bias vector is learned during the course of solving least squares problems on a collection of related tasks. Denevi et al. (2019) extend this approach to a fully online setting and a more general family of linear models.

Previous work investigating the generalisation performance of neural network based on the distance the weights have travelled from their initial values has done so with the aim of explaining why existing methods for training models work well. Bartlett et al. (2017) present a bound derived via covering numbers that shows the generalisation performance of fully connected networks is controlled by the distance the trained weights are from the initial weights. Their bound makes use of a metric that scales with the number of units in the network, which means if the result is extended to a class of simple convolutional networks then the generalisation performance will scale with the resolution of the feature maps. A similar bound can also be proved through the use of PAC-Bayesian analysis (Neyshabur et al., 2018). One can make use of different metrics and techniques for applying

covering numbers to bounding generalisation that do not have the same implicit dependence on the number of units, but they instead depend directly on the number of weights in the network (Long and Sedghi, 2019). Neyshabur et al. (2019) investigate the performance of two layer neural networks with ReLU activation functions, demonstrating that as the size of the hidden layer increases, the Frobenius distance (i.e., the Frobenius norm of the difference between initial and trained weight matrices) shrinks. Inspired by this observation, they show that one can construct a bound on the Rademacher complexity of this class by using Euclidean distance between the initial weights and trained weights of each individual unit. Although they bound the Rademacher complexity directly, they still incur an explicit dependence on the size of the hidden layer, and their analysis is restricted to fully connected networks with only a single hidden layer.

In contrast to these previous studies, our focus is on designing an algorithm that will improve the performance of fine-tuned networks, rather than explaining the performance of the standard fine-tuning methods that are already widespread. Therefore, we do not aim to infer what properties of existing methods enable networks to generalise well—we instead derive such properties and then develop algorithms that enforce them. Moreover, we put a particular emphasis on choosing a metric that is suitable for contemporary convolutional networks, and thus will not scale with the size of the feature maps, while also being easy to implement efficiently.

## 3 DISTANCE-BASED GENERALISATION BOUNDS

Throughout this section we will analyse a loss class consisting of feed-forward neural networks, $f(\vec{x}) = (\phi_L \circ ... \circ \phi_1)(\vec{x})$, where each $\phi_j(\vec{x}) = \varphi(W_j \vec{x})$ is a layer with a 1-Lipschitz activation function, $\varphi$. Both the norm of the weight matrix of each layer and the distance of the fine-tuned weights from the pre-trained weights are bounded from above. Formally, we define

$$\mathcal{F}_* = \{(\vec{x}, y) \mapsto l(y, f(\vec{x})) : \|W_j\|_* \leq B_j^*, \|W_j^0\|_* \leq B_j^*, \|W_j - W_j^0\|_* \leq D_j^*\},$$

where $j$ goes from 1 to $L$, $l$ is a $\rho$-Lipschitz loss function with a range of $[0, 1]$, and $W_j^0$ is the pre-trained weight matrix for layer $j$. Each of the pre-trained weight matrices can be either random variables drawn from a distribution with support such that the constraints are always fulfilled, or they can be fixed (i.e., drawn from a Dirac delta distribution). The only other requirement is that they are independent of the training data used for fine-tuning. From an empirical perspective it is most useful to consider them non-random quantities. We have used $*$ as a placeholder for the norm used to measure the magnitude of weight matrices and the distance from the pre-trained weights. The primary focus is on using the MARS norm,

$$\|W\|_\infty = \max_j \sum_{i=1} |W_{j,i}|,$$

to prove a bound on the empirical Rademacher complexity (Bartlett and Mendelson, 2002) of $\mathcal{F}_\infty$. Our main theoretical result, presented below, is obtained by modifying the "peeling"-style arguments typically used to directly prove bounds on the Rademacher complexity of neural network hypothesis classes (for examples, see Neyshabur et al. (2015); Golowich et al. (2018)). Our modification of the argument allows us to rephrase the resulting theorem in terms of the distance the parameters can move from their initialisation during training.

**Theorem 1.** *For all $\delta \in (0, 1)$, the expected loss of all models in $\mathcal{F}_\infty$ is, with probability $1 - \delta$, bounded by*

$$\mathbb{E}_{(\vec{x}, y)}[l(f(\vec{x}), y)] \leq \frac{1}{m} \sum_{i=1}^m l(f(\vec{x}_i), y_i) + \frac{4\sqrt{\log(2d)} c \rho C_\infty \sum_{j=1}^L \frac{D_j^\infty}{B_j^\infty} \prod_{j=1}^L 2B_j^\infty}{\sqrt{m}} + 3\sqrt{\frac{\log(2/\delta)}{2m}},$$

*where $m$ is the number of training examples, $c$ is the number of classes, $\vec{x}_i \in \mathbb{R}^d$, and $\|\vec{x}_i\|_\infty \leq C_\infty$.*

Crucially, when $l$ is chosen carefully (e.g., the ramp loss), the expectation in Theorem 1 is an upper bound for the expected classification error rate. The proof for Theorem 1 can be found in the supplemental material. The two main terms in this theorem are (i) the product of bounds on the layer norms; and (ii) the summation, which is a bound on the distance the fine-tuned weights can be from the pre-trained weights. The first of these is primarily dependent on the weights obtained via pre-training, whereas (ii) can be controlled during the fine-tuning process. Moreover, one would expect that if better initial weights are selected via pre-training, then the distance the final weights will be from the initial values will be smaller, thus leading to better generalisation. This is the motivation behind the regularisers we develop in Section 4.

An analogous bound is also derived for the Frobenius norm (i.e., $\mathcal{F}_F$) to facilitate a theoretical comparison between the proposed regularisation method and $\ell^2$-SP, an existing approach that relies on penalising Euclidean distance between the pre-trained and fine-tuned parameters. In order to avoid a direct dependence on the number of parameters (as accomplished in Theorem 1) it is necessary to restrict $\mathcal{F}_F$ to use only ReLU activation functions.

**Theorem 2.** *For all $\delta \in (0,1)$, the expected loss of all models in $\mathcal{F}_F$ is, with probability $1 - \delta$, bounded by*

$$\underset{(\vec{x},y)}{\mathbb{E}}\left[l(f(\vec{x}),y)\right] \leq \frac{1}{m}\sum_{i=1}^{m} l(f(\vec{x}_i),y_i) + \frac{2\sqrt{2}c\rho C_2 \sum_{j=1}^{L} \frac{D_j^F}{2B_j^F \prod_{i=1}^{j}\sqrt{n_i}} \prod_{j=1}^{L} 2B_j^F \sqrt{n_j}}{\sqrt{m}} + 3\sqrt{\frac{\log(2/\delta)}{2m}},$$

*where $m$ is the number of training examples, $c$ is the number of classes, $\|\vec{x}_i\|_2 \leq C_2$, $\varphi(\cdot) = \text{ReLU}(\cdot)$, and $n_j$ is the number of columns in $W_j$.*

We can make several observations by comparing our two bounds with each other and previously published bounds: (i) in contrast to Theorem 1, Theorem 2 incurs a significant dependence on the resolution of the intermediate feature maps due to the $\sqrt{n_j}$ factors; (ii) each term in the summations in our bounds can be at most one, whereas the corresponding terms in Bartlett et al. (2017) can be more than one due to the use of multiple types of norms; (iii) our bound from Theorem 1 does not have a direct dependence on the number of weights, in contrast to the bound provided by Long and Sedghi (2019). More detailed comparisons can be found in the supplemental material.

## 4 FINE-TUNING WITH DISTANCE REGULARISATION

The analysis presented in Section 3 suggests that the weights in fine-tuned models should be close to the pre-trained weights in order to achieve good generalisation performance. More specifically, the learning process should search only within a set of weights within a predefined distance from the pre-trained weights. We discuss two strategies for accomplishing this: using projected subgradient methods to enforce a hard constraint, and augmenting the standard cross entropy objective with a term that penalises the distance between the pre-trained and fine-tuned weights. The method based on projection functions is attractive because it guarantees that the constraints will be fulfilled even if one uses a heuristic optimisation method to train the network parameters. However, the penalty-based approaches are more common in the literature, and convenient from an implementation point of view due to the ubiquity of automatic differentiation. Nevertheless, in contrast to the projection-based methods, the techniques that use penalties have weaker assurances on whether a constraint is actually being enforced. We discuss the drawbacks of this non-equivalence further in the supplemental material.

### 4.1 OPTIMISING WITH PROJECTIONS

One way to enforce constraints on the weights of neural networks during training is to use a variant of the projected stochastic subgradient method. This is similar to typical stochastic subgradient methods used when training neural networks, but has the additional step of applying a projection operation after each weight update to ensure that the new weights lie inside the set of feasible parameters that satisfy the constraints. In the case of classic subgradient descent, in order to guarantee convergence towards a stationary point the projection function must perform a Euclidean projection,

$$\pi(\widehat{W}) = \underset{W}{\arg\min} \frac{1}{2}\|W - \widehat{W}\|_2^2$$
$$\text{s.t. } g(W) \leq 0, \tag{1}$$

where $\widehat{W}$ are the newly updated parameters that may violate the constraint, $W$ are the projected parameters, and $g(\cdot)$ is a convex function specifying the constraint. Although the Euclidean projection is required for the classic projected subgradient method, other optimisation algorithms may require the projection to be performed with respect to a different metric. Looking at different optimisers as instantiations of mirror descent with different Bregman divergences is one way to determine the type of projection that should be performed. Unfortunately some of the most common optimisers used in deep learning, such as Adam (Kingma and Ba, 2015), are not guaranteed to converge even when there are no constraints on the parameters being optimised (Reddi et al., 2018). This makes extending it to perform constrained optimisation a purely heuristic endeavour, further compounded by the fact

that it is also not clear which metric the projection should be performed with respect to. As such, the projections used in our approaches are performed with respect to the norm that is most convenient from an efficiency point of view.

Rather than attempting to constrain the distance between the all pre-trained and fine-tuned weights in the network using a single projection, constraints are applied on a layer-wise basis. This makes optimisation more manageable and also allows practitioners to favour fine-tuning certain parts of the network—e.g., if one is fine-tuning a network pre-trained on photos to perform a task on paintings where the same underlying classes are present, one might wish to allocate more fine-tuning capacity to earlier layers in the network. The types of constraints that we wish to enforce take the form

$$\|W_j - W_j^{(0)}\|_* \le \gamma_j,$$

where $\gamma_j$ is a hyperparameter that corresponds to the maximum allowable distance between the pre-trained weights and the fine-tuned weights for layer $j$. With a minor rearrangement, this yields a constraint specification in the form required for Equation 1,

$$g_j^*(W_j^{(0)}, W_j, \gamma_j) = \|W_j - W_j^{(0)}\|_* - \gamma_j,$$

where we have made the dependence on the pre-trained weights and the hyperparameter explicit. The resulting optimisation problem is

$$\min_{W_{1:L}} \sum_{i=1}^{m} l(y_i, (\phi_L \circ ... \circ \phi_1)(\vec{x}_i)) \tag{2}$$

$$\|W_j - W_j^{(0)}\|_* \le \gamma_j \quad \forall j \in \{1 ... L\}.$$

The remainder of this section presents derivations for the projection functions, $\pi_F$ and $\pi_\infty$, corresponding to this constraint specification when it is instantiated with the Frobenius norm and the MARS norm, respectively. We provide pseudocode in the supplementary material that illustrates how these projections are integrated to the neural network fine-tuning procedure when using a variant of the stochastic subgradient method. We refer to the Frobenius norm instantiation as $\ell^2$-PGM and the MARS norm version as MARS-PGM, where the PGM indicates the use of projection gradient methods.

### 4.1.1 CONSTRAINING FROBENIUS DISTANCE

When using the Frobenius distance, Equation 1 can be rewritten as

$$\pi_F(W^{(0)}, \widehat{W}, \gamma) = \arg\min_W \frac{1}{2}\|W - \widehat{W}\|_F^2$$

$$\text{s.t. } \|W - W^{(0)}\|_F - \gamma \le 0.$$

To simplify the problem, we can instead work on a translated version of the same parameter space where $W^{(0)}$ is the origin. Setting $\widehat{T} = \widehat{W} - W^{(0)}$ and $T = W - W^{(0)}$, the problem becomes

$$\pi_2(\widehat{T}, \gamma) = \arg\min_T \frac{1}{2}\|T - \widehat{T}\|_F^2$$

$$\text{s.t.}\|T\|_F - \gamma \le 0,$$

which is the Euclidean projection onto the $\ell^2$ ball with radius $\gamma$, and has the known closed form solution

$$\pi_2(\widehat{T}, \gamma) = \frac{1}{\max\left(1, \frac{\|\widehat{T}\|_F}{\gamma}\right)}\widehat{T}.$$

Expanding the definition of $\widehat{T}$ and translating back into the correct parameter space yields the Frobenius distance projection function,

$$\pi_F(W^{(0)}, \widehat{W}, \gamma) = W^{(0)} + \frac{1}{\max\left(1, \frac{\|\widehat{W} - W^{(0)}\|_F}{\gamma}\right)}(\widehat{W} - W^{(0)}). \tag{3}$$

### 4.1.2 CONSTRAINING MARS DISTANCE

The constraint on the MARS distance can be equivalently expressed as a collection of constraints on the $\ell^1$ distance of each row in the weight matrix from the corresponding row in the pre-trained weight matrix. That is,

$$\|W - W^{(0)}\|_\infty \le \gamma \iff \|\vec{w}_i - \vec{w}_i^{(0)}\|_1 \le \gamma \quad \forall i,$$

where $\vec{w}_i$ is the $i$th row of $W$. One can then make use of the same translation trick used to derive the Frobenius distance projection function to change the $\ell^1$ distance constraints to $\ell^1$ norm constraints,

$$\pi_1(\widehat{\vec{t}_i}, \gamma) = \arg\min_{\vec{t}_i} \frac{1}{2}\|\vec{t}_i - \widehat{\vec{t}_i}\|_2^2 \tag{4}$$

$$\text{s.t. } \|\vec{t}_i\|_1 - \gamma \le 0,$$

where $\vec{t} = \vec{w}_i - \vec{w}_i^{(0)}$. The problem in Equation 4 is a Euclidean projection onto the $\ell^1$ ball with radius $\gamma$, for which there is no known closed form solution. There exist algorithms to find the $\ell^1$ projection in time linearly proportional to the dimensionality of the vector (Duchi et al., 2008), but they are not amenable to implementation on graphics processing units due to the sequential nature of the computations involved. Instead, we apply a projection that minimises the $\ell^1$ distance between the original point and its projection, subject to the projected point lying inside the $\ell^1$ ball with radius $\gamma$,

$$\pi_1(\widehat{\vec{t}_i}, \gamma) = \frac{1}{\max(1, \frac{\|\widehat{\vec{t}_i}\|_1}{\gamma})}\widehat{\vec{t}_i}.$$

This projection, while not providing the closest feasible point measured in Euclidean distance, still provides a point that satisfies the constraints, but is trivial to implement efficiently. Finally, the projection function for the entire weight matrix is given by applying $\pi_1$ row-wise, and translating back into the correct parameter space,

$$\pi_\infty(W^{(0)}, \widehat{W}, \gamma) = \begin{bmatrix} \pi_1(\widehat{\vec{w}_1} - \vec{w}_1^{(0)}, \gamma) + \vec{w}_1^{(0)} \\ \vdots \\ \pi_1(\widehat{\vec{w}_n} - \vec{w}_n^{(0)}, \gamma) + \vec{w}_n^{(0)} \end{bmatrix},$$

where $\widehat{W}$ contains $n$ rows.

## 4.2 PENALTY METHODS

One popular approach in the literature to encourage a model to not move too far from a set of initial weights is to augment the loss function with a penalty term. In our case, this would involve taking the standard objective for the problem at hand (e.g., cross entropy or the hinge loss), and adding penalty terms corresponding to each layer,

$$\min_{W_{1:L}} \sum_{i=1}^m l(y_i, f(\vec{x}_i)) + \sum_{j=1}^L \lambda_j\|W_j - W_j^{(0)}\|_*, \tag{5}$$

where $\lambda_j$ are hyperparameters used to balance the regularisation terms with the main loss function. Due to the subdifferentiability of the two norms considered in this paper, instantiations of Equation 5 can be trained via automatic differentiation and a variant of the stochastic subgradient method. For the Frobenius norm, we actually penalise the squared Frobenius norm, which recovers the $\ell^2$-SP approach of Li et al. (2018). We refer to the instantiation that penalises the MARS distance as MARS-SP.

## 5 EXPERIMENTS

This section provides an empirical investigation into the predictive performance of the proposed methods relative to existing approaches for regularising fine-tuning, and also conducts experiments to demonstrate which properties of the novel algorithms are responsible for the change in performance. Two network architectures are used: ResNet-101 (He et al., 2016), which is representative of a typical large neural network, and EfficientNetB0 (Tan and Le, 2019), a leading architecture intended for use on mobile devices. Both networks are pre-trained on the 2012 ImageNet Large Scale Visual Recognition Challenge dataset (Russakovsky et al., 2015). The Adam optimiser is used for all experiments (Kingma and Ba, 2015). Information regarding the datasets and hyperparameter optimisation procedure can be found in the supplemental material.

## 5.1 PREDICTIVE PERFORMANCE

The first set of experiments are a performance comparison of the proposed methods and existing regularisation approaches for fine-tuning. The baselines considered are standard fine-tuning with no specialised regularisation, $\ell^2$-SP (Li et al., 2018), DELTA (Li et al., 2019), and label smoothing (LS)

Table 1: Results obtained with different regularisation approaches when fine-tuning ResNet-101 (top) and EfficientNetB0 (bottom) models pre-trained on the ILSVRC-2012 subset of ImageNet. We report the mean ± std. dev. of accuracy measured across five different random seeds.

| Regularisation | Aircraft | Butterfly | Flowers | Pets | PubFig | DTD | Caltech | Avg. Rank |
|---|---|---|---|---|---|---|---|---|
| None | 51.81±0.87 | 70.02±0.16 | 76.68±1.07 | 84.19±0.34 | 75.36±0.67 | 66.15±0.55 | 75.32±0.61 | 6.71 |
| DELTA (Li et al., 2019) | 60.38±1.26 | 77.91±0.24 | 86.57±0.27 | 88.11±0.52 | 82.23±2.48 | 69.38±0.68 | 78.88±0.27 | 4.14 |
| LS (Müller et al., 2019) | 60.86±0.65 | 76.52±0.23 | 86.49±0.38 | **90.50±0.31** | 84.38±0.47 | 68.08±0.81 | **80.04±0.10** | 3.14 |
| $\ell^2$-SP (Li et al., 2018) | 60.16±1.33 | 76.56±0.19 | 83.11±0.27 | 86.23±0.41 | 83.79±3.69 | 69.87±0.24 | 79.94±0.17 | 4.00 |
| MARS-SP | 58.52±1.23 | 69.54±0.36 | 75.90±1.04 | 84.22±0.45 | 79.17±0.73 | 69.53±0.93 | 79.19±0.33 | 5.57 |
| $\ell^2$-PGM | 69.61±0.15 | 77.97±0.13 | 86.91±0.66 | 90.47±0.33 | 84.14±0.81 | **70.97±0.71** | 78.45±0.13 | 2.57 |
| MARS-PGM | **72.04±0.70** | **79.50±0.36** | **87.42±0.41** | 89.23±1.36 | **88.75±0.28** | 70.23±0.53 | 79.11±0.19 | **1.86** |
| Regularisation | Aircraft | Butterfly | Flowers | Pets | PubFig | DTD | Caltech | Avg. Rank |
| None | 54.58±0.65 | 69.43±0.47 | 77.43±0.14 | 84.87±0.19 | 75.51±0.83 | 64.98±0.43 | 81.07±0.38 | 7.00 |
| DELTA (Li et al., 2019) | 70.61±0.18 | 79.61±0.21 | 84.60±0.23 | 89.27±0.22 | 88.62±0.54 | 71.37±0.35 | 81.53±0.38 | 3.57 |
| LS (Müller et al., 2019) | 56.14±0.27 | 71.52±0.14 | 83.42±0.70 | 84.95±0.55 | 77.05±0.31 | 65.53±0.26 | 83.12±0.17 | 5.43 |
| $\ell^2$-SP (Li et al., 2018) | 69.26±0.26 | 78.88±0.27 | 86.61±0.48 | 89.79±0.28 | 84.04±0.98 | 71.12±0.52 | 82.91±0.39 | 3.71 |
| MARS-SP | 66.96±0.49 | 72.01±0.20 | 77.79±0.48 | 89.24±0.40 | 85.33±0.59 | 69.41±0.49 | 82.10±0.24 | 5.0 |
| $\ell^2$-PGM | 70.87±0.33 | 81.81±0.22 | 86.95±0.17 | 89.33±0.19 | 88.45±0.36 | 71.63±0.59 | 84.30±0.09 | 2.29 |
| MARS-PGM | **75.22±0.34** | **82.32±0.10** | **90.36±0.15** | **91.38±0.24** | **90.30±0.40** | **73.57±0.38** | **84.84±0.17** | **1.00** |

as formalised in Müller et al. (2019). Once hyperparameters are obtained, each network architecture and regulariser combination is fine-tuned on both the training and validation folds of each dataset. The fine-tuning process is repeated five times with different random seeds to measure the robustness of each method to the composition of minibatches and initialisation of the final linear layer, which is trained from scratch. Test set accuracy is reported for the ResNet-101 and EfficientNetB0 architectures in Table 1.

Comparing the average ranks of the methods across different datasets (Demšar, 2006), the most salient trend is that the projection-based methods exhibit a significant increase in accuracy over their penalty counterparts that use the same distance metrics. This suggests that, in the case of fine-tuning, using a projection to enforce a constraint on the weights throughout the training process is a better regularisation strategy than adding a term to the objective function that penalises the deviations of weights from their pre-trained values. Additionally, looking further at the relative performance of the two projection-based regularisers, we can see that the MARS distance variant is more often a better choice than the Frobenius distance. This observation further supports the conclusions of the comparison of the bounds in Section 3.

## 5.2 DISTANCE FROM INITIALISATION

To further investigate the relationship between the penalty and projection strategies for regularisation, we analyse the distances between the pre-trained and fine-tuned weights at a per-layer basis. Figure 1 provides histograms indicating the distributions of per-layer MARS distances for the models without any regularisation, regularised with MARS-SP, trained with the MARS-PGM, and also DELTA. All networks were trained on the Pets dataset with the same hyperparameters used by the models examined in Section 5.1. We observe three trends from these plots. Firstly, both the penalty and projection methods reduce the MARS distance between the pre-trained and fine-tuned parameters relative to the unregularised model. Secondly, a significant number of the constraints enforced by the projection method are activated—i.e., many of the weight matrices lie on the boundary of the feasible set. In contrast, the penalty-based regulariser does not enforce similarly activated constraints. Finally, the results demonstrate that the DELTA method of Li et al. (2019) does not operate by implicitly regularising the same quantity, as its MARS distance histogram is longer tailed than the others.

## 5.3 CAPACITY CONTROL

To demonstrate the ability of the distance-based regularisation methods to control model capacity, we sweep through a range of hyperparameter values and plot the corresponding predictive performance. Hyperparameters were generated according to $\lambda_j = c\hat{\lambda}_j$ and $\gamma_j = c\hat{\gamma}_j$, where $c$ is varied, and $\hat{\lambda}_j, \hat{\gamma}_j$ are the values found during the hyperparameter optimisation process. Plots of $c$ versus the resulting accuracy on the pets dataset are given in Figure 2 for both the ResNet101 and EfficientNetB0 architectures. We can see that the PGM methods behave as the theoretical analysis predicts: hyperparameter

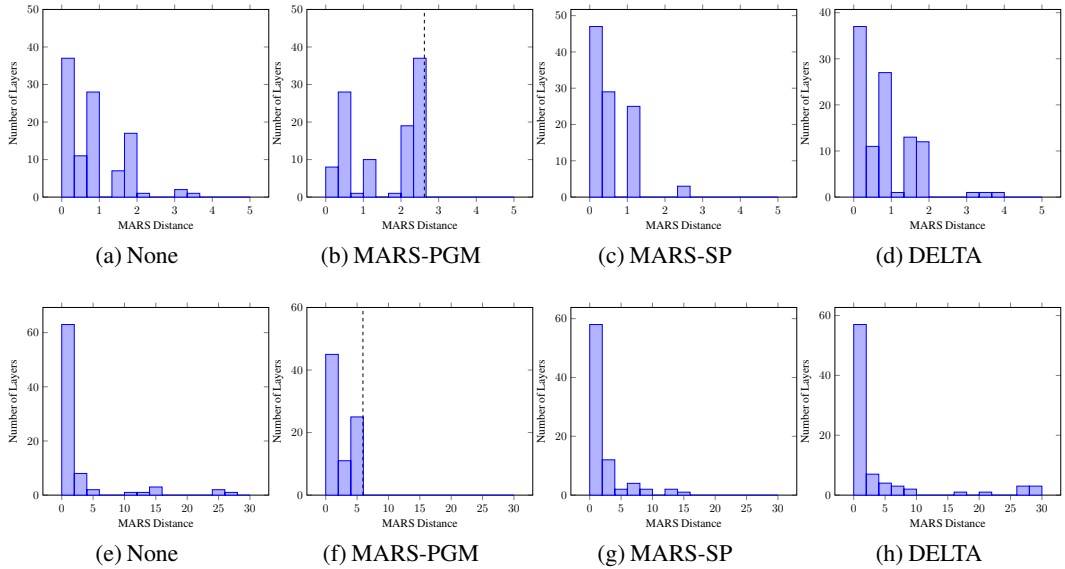

Figure 1: Histograms of MARS distances between pre-trained and pets dataset fine-tuned weights for the specified regularisation strategies. MARS-PGM successfully constrains weight distances to be less than $\gamma_j$, indicated by the dashed line. Top: ResNet101, Bottom: EfficientNetB0.

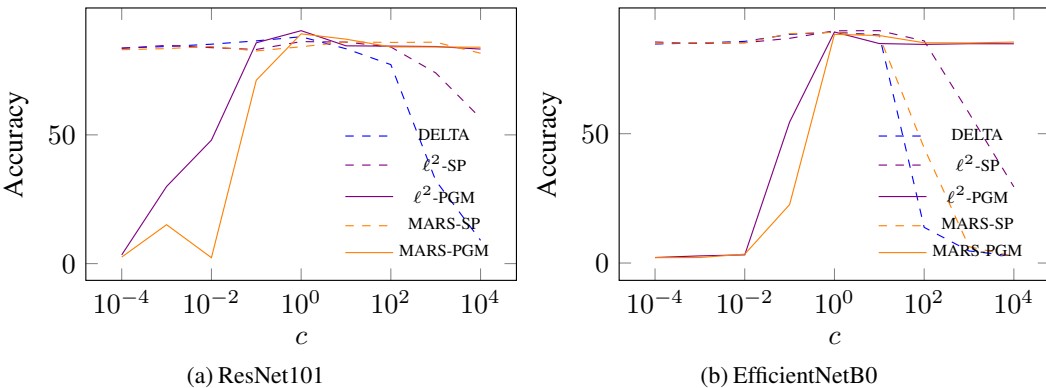

Figure 2: Sensitivity of the regularisation methods to the choice of hyperparameters. Measurements are taken on the pets dataset, and $c$ is a factor applied to the hyperparameters found during tuning.

configurations that lead to very small distances between pre-trained and fine-tuned weights result in underfitting, relaxing the hyperparameters too much leads to overfitting, and using the optimised hyperparameters (i.e., when $c = 1$) achieves the best performance.

### 5.4 EMPIRICAL COMPARISON OF BOUNDS

We perform an empirical comparison of our two bounds, along with a bound based on the spectral norm (Long and Sedghi, 2019), to demonstrate the relative tightness. This is done by training neural networks on the MNIST dataset (LeCun et al., 1998). We use the architecture of Scherer et al. (2010), which consists of a single convolutional layer with $9 \times 9$ filters and 112 channels, followed by $5 \times 5$ max pooling, and finally a linear classifier layer. The network is trained for 15 epochs using the Adam optimiser (Kingma and Ba, 2015), as training any further does not result in any performance increase. Following similar work that has performed empirical evaluation of neural network bounds, we evaluate the bounds by computing the relevant norms and distances of the trained network weights. These quantities are then used in place of the upper bounds (i.e., $B_i^*$ and $D_i^*$) used in the definition

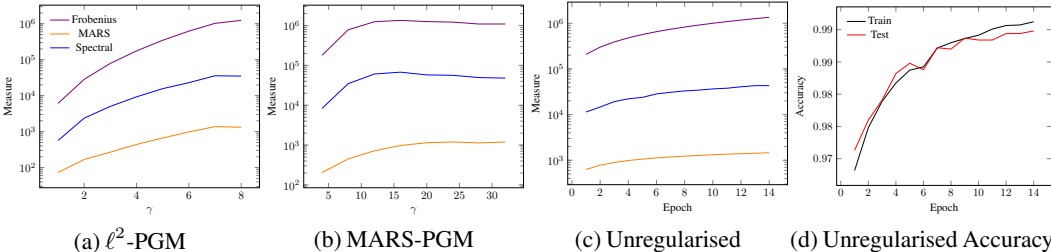

(a) $\ell^2$-PGM      (b) MARS-PGM      (c) Unregularised      (d) Unregularised Accuracy

Figure 3: An empirical comparison of the tightness of our bounds on the Rademacher complexity, and that of Long and Sedghi (2019). The (a) and (b) plots demonstrate the empirical value of the bounds as a function of the regularisation parameters: the vertical axes correspond to the model complexities for each of the three measures considered, and the horizontal axes represents the regularisation strength for $\ell^2$-PGM (a) and MARS-PGM (b). The plot in (c) shows how the model complexity measures change throughout the training of an unregularised model, and (d) shows the training and test set performance throughout training for the same unregularised model.

of the hypothesis classes considered by Theorems 1 and 2. In keeping with previous work that has performed this type of empirical comparison, we measure the distance from random initialisation, rather than fine-tuning from a pre-trained network (Bartlett et al., 2017; Neyshabur et al., 2019). Figure 3 shows how these three quantities vary for different choices of $\gamma$ (the same hyperparameter is used for both layers) when using MARS-PGM and $\ell^2$-PGM, and also how they differ through the process of training and unregularised model. We observe several trends: (i) like previous work in this area, all of the empirically evaluated Rademacher complexity bounds are still too loose to be useful for model selection and providing performance guarantees on the expected test set performance; (ii) the bounds based on the MARS norm are consistently tighter (by orders of magnitude) than the bounds based on the Frobenius and spectral norms; (iii) the empirical measurements of model complexity plateau even though the hyperparameters governing the worst-case capacity continue to increase—we suspect this is caused by implicit regularisation from early stopping. This implicit regularisation from early stopping also provides an explanation for why we observe only a small degradation in performance in Figure 2 when the hyperparameters are set to very large values.

## 6    CONCLUSION

This paper investigates different regularisation methods for fine-tuning deep learning networks. To facilitate this, we provide two new bounds on the generalisation performance of neural networks based on the distance of the final weights from their initial values. The discussion comparing these bounds suggests that the MARS distance is a more appropriate metric in the parameter space of convolutional networks than Frobenius distance. Additionally, several new algorithms are presented that enable an experimental comparison between different regularisation strategies. The empirical results corroborate our theoretical investigation, demonstrating that constraining MARS distance is more effective than constraining Euclidean distance. Crucially, we also show that, in line with our theoretical results, enforcing a hard constraint throughout the entire training process on the distances the parameters can move is far more effective than the widely used strategy of adding a penalty term to the objective function. Implementations of the methods used in this paper are available online.[1]

## ACKNOWLEDGMENTS

This work was supported by the Engineering and Physical Sciences Research Council (EPSRC) Grant number EP/S000631/1; and the MOD University Defence Research Collaboration (UDRC) in Signal Processing.

---

[1] https://github.com/henrygouk/mars-finetuning

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
