# OpenReview forum: "Distance-Based Regularisation of Deep Networks for Fine-Tuning"
_ICLR.cc/2021/Conference — ICLR 2021 Poster_

### Official Review · AnonReviewer4 · 2020-10-26
**Interesting for both theory and practice, although the connection between the two is a bit weak**

**Rating:** 7
**Confidence:** 4

**Review:**

In this manuscript the authors derive a bound on the rademacher complexity of neural network models which can be written as a funciton of the MARS norms of the weights in the network. This motivates the authors to put a regularization on the MARS norm of the network weights instead of the more typical L2 norm. Here the authors implement this regularization as a hard bound on the weights, which they enforce by projecting the weights back on the allowed ball. They use their regularization for transfer learning of ResNet-101 and EfficientNetB0 from ImageNet onto the set of smaller image classification tasks. On these tasks, the projection methods and to a smaller degree the MARS based methods generalize better.

Overall I vote for acceptance. This is an interesting contribution to the literature, providing both a theoretical insight and an experimental test that this theoretical insight is relevant for applications. However there is a certain disconnect between the theory and the experimental observations. Performance  benefits more from the projection methods than from the switch of norm although the switch of norm has a much stronger theoretical justification.

Pros:
1) Well structured paper with interesting results
2) Theoretical results are well justified to be more helpful than existing bounds.
3) There is an empirical test that the switch in bound is helpful for practice.
4) Overall the generalization is actually improved.

Cons:
1) Empirically the less justified change has a larger impact, indicating that there might be another more important theoretical insight
2) The hyperparameter setting procedure remains opaque. The authors always talk about gamma_i/ lambda_i parameters changing the strength of regularization per layer, but only test how scaling all regularizations up or down affects performance. A description how the values were chosen is really necessary I think and some analysis to convince us that the worse performance of the regularization is not caused by a bad hyperparameter choice would definitely be a plus.
3) I think there is a bit of a missed opportunity here for the scaling over layers as the bound suggests an unequal weighting of the layer norms. I think directly regularization of the bound which would allow layers to compensate for each other or giving each layer an equal budget in terms of raising the bound would be interesting variants here.

---

> ### Author Response · Authors · 2020-11-20
> **Response to Reviewer 4**
>
> Thank you for your positive review!
>
> ### Q1: Justification of penalty vs projection
> A1: We agree that moving from a penalty to a constraint has the larger impact of the two changes, but we still feel that our justification for why this occurs is strong: the only conditions in which penalty-based regularisers have been shown to be equivalent to constraint-based methods (e.g., by Oneto et al.) are not fulfilled when training neural networks. I.e., the common belief that penalties are equivalent to a norm constraint when training neural networks has never actually been justified.
>
> ### Q2: Hyperparameter Setting.
> A2: Additional details regarding hyperparameter tuning are given in Appendix F. Specifically, we use a Bayesian optimisation approach consisting of the tree of Parzen estimators method implemented the HyperOpt framework. Each dataset/method/architecture combination is given 20 interations of hyperparameter optimisaiton. The `hptune.py` file in the supplemental material contains the implementation for this.
>
> ### Q3: Unequal weighting suggestion.
> A3: Thanks for the great suggestion. We agree that this is interesting to do, and we are actually already pursuing this as part of an extension to this method!

---

### Official Review · AnonReviewer1 · 2020-10-28
**the proof seems problematic**

**Rating:** 6
**Confidence:** 4

**Review:**

The work proposes a Rademacher type bound for the fine-tuned models based on the distance between the fine-tuned weights and the pre-trained weights. Since the distance term shows up in the upper bound on the generalization gap, the authors further propose to adopt it as the regularization term to boost the generalization performance of the model during the fine-tuning process. Some experiments are also done to show the effectiveness of the proposed regularization.

I am seriously concerned about the correctness of the Rademacher-type bound the authors have proposed. The bound does not seem correct to me.

The flaw comes from the function class F_* defined in section 3 of the draft. The function class F_*, by definition, depends on the pre-trained weights W_j^0. However, W_j^0 is not fixed, it is random! This is because W_j^0 depends on the data (W_j^0 is pre-trained using the data), which by the assumption of the draft, is random. As a consequence you cannot assume W_j^0 as fixed. The randomness of the hypothesis class F_* destroys almost all the derivations the authors are currently using in their proof.

Another minor bug is the second term in the bound for theorem 1 seems to have some subscript issues. To me the product term related to B_j^\infty should go from i=1 to j instead of from j=1 to L. I may have missed something in this point but could the authors double check if the subscript of the B_j^\infty is correct? In particular the derivation from the second inequality to the third on page 13 of the appendix.

The second issue is easy to fix. However the first issue seems like a fundamental flaw. I do not have a good way to handle it for now.

---

> ### Author Response · Authors · 2020-11-20
> **Respones to Reviewer 1**
>
> ### Q1: Correctness of the Proof?
> A1: The proof is correct. We emphasize that, contrary to the reviewer's assumption, the pretrained weights are *independent of the training data used for fine-tuning*, and hence they are *not* random variables. Therefore the proof holds. To elaborate, we operate within the typical deep learning paradigm, where models are pre-trained on large auxiliary datasets, such as ImageNet. This is done independently to, and in advance of, fine-tuning on target dataset whose generalisation properties we are analysing. Several other learning theory papers, also used this construct of a non-random/fixed initial condition. For example, the cited Denevi NeurIPS'18 and Denevi ICML'19 papers use this idea for linear models. We use this idea for deep network models.
>
> ### Q2: Proof subscript?
> A2: This subscript is correct, but we agree the presentation of this part of the proof could be improved. Our updated proof will make the reason more obvious.
>
> We hope that we have clarified the reviewer's main issue with the paper as being due to a small misunderstanding. We are happy to answer any further questions you may have about the paper now that this is cleared up.

---

> > ### Comment · AnonReviewer1 · 2020-11-21
> > **my concerns on Q1 are not fully addressed**
> >
> > Thanks very much for the clarification. In particular I would like to thank the authors for double checking the subscripts for Q2.
> >
> > Somewhat the authors’ comment on Q1 does not fully address my concern.
> >
> > Pretrained weights are independent of the training data used for fine-tuning, but that does not lead to the conclusion that they are not random. They can still be random variables and independent of the fine-tuning process. If so what the authors have proved is a generalization bound conditioned on W_0. To give a full claim on the 1-\delta probability you may want to work on a theorem for all W_0s by either a union bound or other types of more delicate methods. Otherwise the probability 1-\delta does not seem right.
> >
> > To me the only way that you can claim W_0 is fixed is by assuming the data in the pretraining process are all fixed. This assumption seems so strong that the problem is not interesting any more.

---

> > > ### Comment · AnonReviewer2 · 2020-11-22
> > > **not sure if Q1 poses any concern**
> > >
> > > Thanks R1 and authors for the discussion.
> > >
> > > I do not understand Q1 completely, however. The authors assume a norm bound on the initial weights as well as a norm bound on the final weights and the distance between initial and final weights. In learning theory these bounds seem to be perfectly in line with standard assumptions in the field.
> > >
> > > Applying a union bound (as R1 suggests) will not further any insight as the results already hold with high probability for *any* initial weights W_0 that satisfy the bounded norm constraint. Moreover, the results also hold for *any* final weights W_t that satisfy their norm constraint and the distance constraint.
> > >
> > > Sure, having a bound that would *simultaneously* be true for all W_0 and W_t within the respective regions can be obtained, and a simple way would be by a covering argument and union bound over the respective balls, leading to an additional covering radius (radii?) in the bound. But what exactly is the utility of such a result?
> > >
> > > Whenever we perform fine-tuning or transfer learning in practice, we are always provided with the initial weights, therefore it does not add much value to have a result that is needlessly diluted to hold simultaneously over all possible initial weights. The essence of Theorem 1 is to provide a control over model capacity that is usable in practice, and the 1-\delta is to account for the randomness in the data, and not the model space itself.
> > >
> > > One can further this line of reasoning and ask for bounds that are averaged over all possible initial weights for the first training procedure, since they also are sampled at random (in fact, it would make more sense there, since those weights are actually initialized randomly). One can also question the norm bound, since we could be given a W_0 that does not obey the norm bound itself, and then we must have to make another covering argument.
> > >
> > > In summary, it boils down to the assumptions that the authors make, and given that every transfer learning algorithm till date assumes knowledge of initial weights (I have yet to see a transfer learning algorithm that uses random initial weights, but I may be wrong), I do not see any reason why that cannot be assumed for this result as well.

---

> > > > ### Comment · AnonReviewer1 · 2020-11-22
> > > > **we are on the same page**
> > > >
> > > > Thanks R2 for the discussion. Actually what I suggested for Q1 was very much aligned with R2’s comments but with a different opinion.
> > > >
> > > > To claim a 1-\delta probability bound the authors can do either of the following:
> > > >
> > > > 1). take into account the randomness of W_0 by averaging all possible W_0 of the pre-training procedure. I suggested a union bound but that is an overkill. looks like the authors are not going this route but to me this may be the desired way.
> > > >
> > > > 2). assume W_0 is fixed. With a fixed W_0 one can follow the normal Rademacher calculus to get some generalization guarantee. However this is implicitly assuming that a) the pre-training data are all fixed. b) the pre-training procedure is deterministic. This is usually NOT the case in the pre-training process.
> > > >
> > > > As R2 says, it all boils down to the assumptions the authors have made. To me the assumption that W_0 is fixed is too strong and the problem is not interesting any more.  I would prefer the authors do 1) in their theorem instead.

---

> > > ### Author Response · Authors · 2020-11-22
> > > **Does not seem to be an issue**
> > >
> > > I guess you are talking about bounding, with high confidence, the Rademacher complexity of a random hypothesis class (c.f., deterministic hypothesis class if initial weights, $\mathcal{W}^0$, are fixed), in the sense that a distribution over initial weights induces a distribution over hypothesis classes. Here is a sketch that shows our bound still works for randomly selected initial weights, provided the initial weights are independent of the fine-tuning dataset.
> > >
> > > Pick any distribution, $Q$, over pre-trained weights, $\mathcal{W}^0$, with support such that the given norm constraints are fulfilled. I.e., we have that $\mathcal{W}^0 \sim Q$, and we have the random hypothesis class $\mathcal{F}_{\mathcal{W}^0}$.
> > >
> > > The key step in the theorem(s) in the manuscript under review is to prove a bound on the empirical Rademacher complexity, $\hat{R}(\mathcal{F}_{\mathcal{W}^0})$, in the case where $\mathcal{W}^0$ is non-random.
> > >
> > > One can extend our theorem(s) to cope with random hypothesis classes by bounding $\sup_{w \in \text{support}(Q)} \hat{R}(\mathcal{F}_w)$, which is the bound achieved when we sample the worst possible pre-trained weights from any distribution that satisfies the norm constraints specified in the paper. That is, we can construct a uniform bound over all relevant hypothesis classes.
> > >
> > > Denote by $T(\mathcal{F}_w)$ the upper bound we obtain for $\hat{R}(\mathcal{F}_w)$ when proving our theorem(s). Because our bound(s) are in terms of the constraints on the norms, and not the norms of the pre-trained weights themselves, we have that $T(\mathcal{F}_a) = T(\mathcal{F}_b) = t$ for all $a,b \in \text{support}(Q)$ and some $t$.
> > >
> > > From this we can conclude that $\sup_{w \in \text{support}(Q)} \hat{R}(\mathcal{F}_w) \leq t$. As a consequence, our current bounds hold even when the pre-trained weights are random.

---

> > > > ### Comment · AnonReviewer1 · 2020-11-23
> > > > **Thanks for the sketch of proof. My concern is addressed**
> > > >
> > > > This is exactly what I was asking for.
> > > >
> > > > So basically the key in your sketch of proof is that the bound only depends on the norm constraints instead of W_0, so it holds uniformly for all W_0s that satisfy the norm constraints.
> > > >
> > > > Can you add a formal proof to the appendix and extend your theorem from a fixed W_0 to the uniform bound?
> > > >
> > > > I am changing my rating to 6 to reflect the change.

---

> > > > > ### Author Response · Authors · 2020-11-23
> > > > > **The existing proofs already work**
> > > > >
> > > > > The existing proofs in our appendices already work when the initial weights are random variables. We will change the definition of $W_i^0$ given in the main text to make it clear that our bounds apply in this situation as well.

---

### Official Review · AnonReviewer3 · 2020-10-28
**Official Blind Review #3**

**Rating:** 5
**Confidence:** 4

**Review:**

This paper proposes new regularization methods for fine-tuning deep neural networks based on matrix $\infty$-norm distance. The authors claim that their choice of matrix $\infty$-norm distance is more suitable than commonly used Frobenius norm distance (a.k.a., Euclidean distance) when measuring the distance in the parameter space of convolutional networks by a comparison of two generalization bounds. Moreover, the authors empirically show that enforcing a hard constraint on the weights by projected methods throughout the training process is more effective in regularizing neural networks than widely used strategy of adding a penalty term to the objective function.

Overall, the paper is well written and has a nice logical flow. The problem of finding a proper distance metric for fine-tuning is interesting, though I have a few concerns outlined below regarding their theoretical analysis of using generalization bound to guide the choice of distance metric, especially the proof of the theorems.

Concerns:
1. The authors try to modify the peeling technique of prior work to prove two generalization bounds, i.e., Theorem 1 and Theorem 2. A key step in proving the two theorems is to prove Lemma 2 given in the Appendix. However, from the proof of Lemma 2, if I understand correctly, the second equality and the fourth equality seem to interchange the order of sum and supremum freely, i.e., $\sum_{j=1}^n v_j \sup_{W_{1:k}} …=\sup_{W_{1:k}} \sum_{j=1}^n v_j…$, which of course does not hold in general. It should be stated clearly on why the two equalities hold here.

2. The authors provide two generalization bounds for fine-tuning. The two bounds are almost the same except for the norm used. The authors then claim that a comparison of the two bounds suggests that matrix $\infty$-norm is more effective than Frobenius norm when measuring distance in weight space of neural networks just because matrix $\infty$-norm itself is independent of the feature map size. This is misleading in the sense that matrix $\infty$-norm and Frobenius norm are actually equivalent, i.e., for an arbitrary matrix, its matrix $\infty$-norm is not strictly smaller than its Frobenius norm and vice versa, and thus the two bounds are also equivalent and cannot be used to tell which norm is better. Therefore, I do not think that their choice of matrix $\infty$-norm  as the distance metric can be theoretically justified by comparing the two generalization bounds as in the paper, despite that empirical results show that their method performs well in practice.

3. In section 5.3, the authors hope to demonstrate the ability of the distance-based regularization methods to control model capacity by sweeping through a range of hyperparameter values and plotting the corresponding predictive performance. The authors claim that Figure 2 shows that the PGM methods behave as the theoretical analysis predicts and the penalty-based approaches are not able to influence the model capacity as much as the constraint based approaches. This statement is inaccurate in several ways. First, the symbol $\lambda_j$ in the third line is confusing. It seems to represent the hyperparameter for both the constraint based methods and penalty methods. However, $\lambda_j$ first appears in equation (5) where it represents the hyperparameter for penalty methods. Second, from Figure 2, as $c$ becomes larger and larger, there is only a very small drop of accuracy for the PGM methods. So, it does not lead to overfitting, and the PGM methods do not behave exactly as the generalization bound predicts. Third, small $c$ for PGM methods corresponds to large $c$ for penalty methods by the equivalence of constraint methods and penalty methods. Therefore, Figure 2 shows that the penalty-based approaches actually have the same influence on the model capacity as the constraint based methods.

Minor comments:
- From the proof of Theorem 2, the term $\sqrt{c}$ in the bound should be $c$. Therefore, the bounds in Theorem 1 and Theorem 2 exhibit the same dependence on the number of classes.

- I am a little confused by the sentence “In the case of the final classification layer, $W_L^0$ can be randomly initialized.” in 7th line of Section 3. Do you mean that $W_j^0$s ($j<L$) are pre-defined and fixed, but $W_L^0$ is random? However, when proving the upper bound for empirical Rademacher complexity, especially the last step where the rightmost term evaluates to zero, it seems that you assume that all these matrices $W_j^0$ ($1\leq j\leq L$) are fixed. It would be better if this can be clarified.

- In section 4 and Appendix E, to support the claim that projection based methods are better than penalty based methods, the authors state that penalty methods have weaker assurance on whether a constraint is being forced. However, Figure 1 shows that for ResNet101 model penalty-based method is actually more effective in enforcing the constraints in the sense that not only it successfully constraints weight distance to be less than $\gamma_j$, but also the number of weights which have small distance is larger. Therefore, more evidence might be needed to support their claim.

Some typos:

(1) In line 6 of Page 2, best way restrict -> best way to restrict

(2) In the last line of Page 5, change the $l^1$ distance-> change the MARS distance

(3) In the third line of the proof of Lemma 2, $\varphi_j$-> $\varphi$

(4) In the third formula of the proof of Theorem 2, $sqrt{2}$-> $\sqrt{2}$

---

> ### Author Response · Authors · 2020-11-20
> **Response to Reviewer 3**
>
> Thank you for the very detailed review!
>
> ### Q1: Proof correctness
> A1: Thank you for finding this---it is a mistake in the original proof. We have changed the proof strategy slightly to overcome this.
>
> ### Q2: Equivalence of norms
> A2: After fixing the proofs the bound for the Frobenius norm class of neural networks has becomes looser. Specifically, it now explicitly depends on the size of the feature maps in each of the intermediate layers, whereas the MARS norm class does not have this dependence. We have changed the discussion in the comparison between the bounds to highlight this.
>
> ### Q3: Model capacity control
> A small drop in accuracy as model capacity increases still consitutes a small amount of overfitting, but we have tempered the claim in the paper to reflect this point. Note that the generalisation bounds are just that---bounds. They predict the worst case not the average case, so it would be erroneous to expect that the plots in Figure 2 follow the exact trend present in the bound.
>
> Regarding the model capacity control of PGM vs SP regularisers, consider the MARS-SP and MARS-PGM hyperparameter senstivities on ResNet-101. When the $\gamma_j$ are moved two orders of magnitude away from their optimal values, performance has degraded to zero due to underfitting (i.e., not enough model capacity). In contrast, moving the $\lambda_j$ values *four* orders of magnitude away from their optimal values results in negligible decrease in performance (i.e., little change in model capacity). A similar, but less pronounced, trend can be observed for the other SP vs PGM comparisons as well. Note that we include DELTA in the plots for completeness, and have not made any strong claims about its model capacity control abilities in the paper.
>
> ### Minor comments:
> * This has been fixed in the latest version.
> * We consider $W_L^0$ a fixed quantity, just like the pre-trained weights in the other layers. Note that even if it was a random variable the proof will still work, as the expectation is over only the Rademacher random variables and we have still defined $W_L^0$ such that there is a bound on its norm.
> * We claim only that the constraint and penalty methods are not equivalent, and that our theory makes sense when a constraint is enforced. One should, of course, still expect that a penalty will do *something*. Our point with this figure is to demonstrate that the two strategies do in fact do something different.

---

> > ### Author Response · Authors · 2020-11-24
> > **RE: Figure 3**
> >
> > After additional discussion with the area chair we have removed the claim comparing hyperparameter sensitivity of the constraint and penalty methods. However the main take-away message of the figure still stands: regularising the Frobenius/MARS distance---whether by a penalty or a constraint---is an effective means for capacity control.

---

> > > ### Comment · AnonReviewer3 · 2020-11-25
> > > **Thanks for the clarification**
> > >
> > > The reviewer thanks the authors for the response. My concerns have been partially addressed. Therefore I have updated my score to reflect the change. Thanks.

---

### Official Review · AnonReviewer2 · 2020-10-29
**Effective, simple and well-motivated, although lacks in comparisons with prior work**

**Rating:** 7
**Confidence:** 4

**Review:**

This paper studies regularization for neural network fine-tuning, motivated by limiting deviation of the final model from the initialization states. The provide a generalization bound that utilizes a novel Rademacher complexity term built on the layer weights and their deviation from the initial weights. This bound relates particularly to fine-tuning, since a part of the bound can be fixed to the pre-trained weights, providing an alternative regularization objective specific to fine-tuning. Using this objective, the authors provide several fine-tuning benchmark experiments and demonstrate competitive performance.

Strengths of the paper:
- Well written, easy to follows.
- Motivation for the algorithm stems directly from the analysis, as opposed to heuristic-style arguments that typically dominate the field of CV /deep learning research, especially for fine-tuning. Moreover, the generalization bounds are derived such that they lead to an optimization objective (as opposed to conventional approaches that typically have not led directly to an effective algorithm).
- The analysis appears to be general, without any particularly strong assumptions.
- Two different norms are considered with corresponding algorithms and experiments.
- Extensive ablations are performed on vision tasks.

Weaknesses:
- Only tested on computer vision benchmarks. If the paper claims this approach to be a general technique then it is necessary that the methods do well on other tasks (e.g., language), otherwise the experimental claims rely too much on the convolutional inductive biases.
- If the paper is in fact framed as a CV paper, then it is natural that a comparison be made with respect to prior (albeit heuristic) computer vision research, e.g., label-smoothing regularization, entropy regularization and so on.
- An empirical comparison of the tightness of the bounds is warranted given the deviation of this analysis from PAC-Bayesian (Neyshabur 2018) or spectral norms (Bartlett and Long).

---

> ### Author Response · Authors · 2020-11-20
> **Response to Reviewer 2**
>
> ### Q1: Benchmarks
> A1: We agree that fine-tuning is an important component in many recent NLP methods, but we are explicit in both the abstract and full paper that we are primarily concerned with convolutional networks. However, the types of architectures used in NLP (namely, transformers and several RNN variants) are quite different to feed-forward neural networks that we theoretically analyse in this paper. Theoretical investigations into the generalisation properties of these other architectures are almost non-existent, so demonstrating what makes these networks generalise would be a significant contribution in its own right. And we leave this to future work.
>
>
> ### Q2: Comparisons.
> A2: Please note that we already compare with two recent methods (L2-SP from ICML 2018 and DELTA from ICLR 2019)  designed for fine-tuning convolutional networks. In our experience label smoothing is typically used during the pre-training phase, rather than the fine-tuning process. That said, we are currently running some label smoothing experiments and will endeavour to update the paper with new results before the end of the discussion phase.
>
>
> ### Q3: Comparison of bound tightness.
> A3: We will add an experiment showing how the bounds compare in practice when updating the paper with the label smoothing experiments.

---

> > ### Author Response · Authors · 2020-11-24
> > **Submission updated**
> >
> > We have updated the submission to include a comparison to label smoothing and an empirical comparison of the model complexity metrics suggested by the theoretical analysis.

---

> > > ### Comment · AnonReviewer2 · 2020-11-25
> > > **thank you for the update**
> > >
> > > Thank you for the update! I have updated my review to reflect accordingly.

---

### Comment · Area_Chair1 · 2020-11-23
**Some more clarifications needed**

Thank you all for the discussion which has already raised and clarified important issues. I found the document rather confusing on some points, so there are still a number of points I would like to raise with all of you (somewhat rewording some of the reviewers' questions):

Focus on fine-tuning:
Why does the paper only discusses fine-tuning? The main theorem provides a bound on the risk without stating anything about the origin of the initial (pre-trained) weights. Thus, this theorem is no less relevant for standard learning with initial random parameters than it is for fine-tuning from pre-trained parameters. Why do the authors only consider fine-tuning, and why is this theorem not related to the ones pertaining to "standard" learning from scratch?

Experiments:
It is quite peculiar to compare algorithms on sub-optimal setups (Table 1) and better setups on flawed experimental benchmarks (Table 3). A simple corrected version of Stanford Dogs is provided in [Li et al.: A baseline regularization scheme for transfer learning with convolutional neural networks. Pattern Recognit. 98 (2020)], avoiding the overlap between the pre-training set of ImageNet and the fine-tuning training set for transfer. I suppose the same protocol could be applied to CUB-2011.

Hyper-parameters:
Still on the subject of experiments experiments, I think that the paper should be more precise regarding the tuning of hyper-parameters, which is crucial here because of the importance given in the paper to the difference between the formulations relying on hard constraints and penalties. Giving the same number of trials in HyperOpt is not sufficient to ensure fairness; the parameters given to HyperOpt should be given, with an assessment of the relevance of the given intervals for gamma and lambda.
Also regarding hyper-parameters, I don't get the message of Section 5.3: what should be inferred from Figure 2 that compares the effects of the variations of the hyper-parameters \gamma and \lambda? These hyper-parameters are not commensurable; why a multiplicative update of the optimal gamma should be expected to correspond to the same (or inverse) multiplicative update of \lambda?

Convergence of penalty-based approaches:
A last point for me very debatable, is the assessment of convergence given in Figure 3. The penalty of MARS-SP being not differentiable, why should the norm of the gradient be relevant for assessing convergence? If the solution is, as expected,  at a non-differentiable point, the norm of all subgradients is not expected to go to zero, and points in the vicinity of the solution are not expected to have small gradients.
Also regarding this figure: the number of epochs reported in the DELTA paper is several thousands, here the experiment shows only 30 epochs: is this representative of all the experiments carried out elsewhere in this paper?

---

> ### Author Response · Authors · 2020-11-23
> **Some answers**
>
> *Q1: Why focus on fine-tuning?*
>
> Yes. We agree the bound can also apply to training from scratch, and we will add some wording on this. The reason we initially focused on fine-tuning is that it is the area where this type of regularisation is more likely to have a stronger positive impact from an empirical perspective. This is because more relevant initial weights will result in less distance needing to be traveled throughout the training process.
>
> *Q2: Choice of benchmarks*
>
> Re: Sub-optimal, could you clarify what you mean by this? We consider our setup in Tab 1 to be well optimized.
> Re: Flawed setups. Thank you for pointing out the 2020 paper and benchmark of Li et al. We are happy to update our experiments to use this version of those benchmarks. That said, it's not clear to us why evaluating on the modified versions of CUB-2011 and Stanford Dogs is more informative than the seven datasets we have already used. The modified version of the benchmarks will also render the results not directly quantitatively comparable to prior work such as DELTA.
>
> *Q3a: Hyperparameter ranges*
>
> We used log-uniform distributions as priors for HyperOpt. In particular, for the PGM methods $\text{ln}(\gamma) \sim \mathcal{U}(0.5, 3.5)$ and for the penalty methods we use $\text{ln}(\lambda) \sim \mathcal{U}(-10, -1)$. These are reasonable settings, because they cover a superset of the hyperparameter range considered by DELTA; and cover the highest performing condition considered by $\ell^2$-SP.
>
> *Q3b: Sec 5.3/Fig2.*
>
> Thanks for pointing this out. We agree that the differences in accuracy are not guaranteed to occur across the same range of $c$ for both constraint and penalty methods. Therefore we will remove the sentence in Sec 5.3 alluding to their direct comparison.
> That said, the main take home message of Fig 2 is to confirm that the constraint hyper-parameter does indeed control model capacity in a manner predicted by the bounds. This observation is still valid.
>
> *Q4: Convergence*
>
> Thanks for pointing this out. We will address this point by changing the plot to monitor the training loss along with the validation accuracy. When using subgradient methods, a common convergence criterion is checking when the change in training loss becomes negligible. In contrast, practitioners usually monitor validation accuracy as a stopping criterion. Therefore the faster convergence of validation accuracy still supports our point that selected models in practice are unlikely be at loss extrema and hence unlikely to benefit from constraint enforcement.
>
> *Q5: Amount of training*
>
> Please note that the plots in DELTA paper have x-axis in units of iterations; while those in our papers have x-axis in units of epochs. So the amount of training is comparable across both papers.

---

> > ### Comment · Area_Chair1 · 2020-11-24
> > **suboptimal experimental setups**
> >
> > Thanks for all the clarifications.
> >
> > To clarify my comment on suboptimal experimental setups, I mean that the learning protocol used here does not lead to the performances obtained elsewhere. For example, the results obtained in the l2-SP and DELTA papers for subsets of Caltech are much better than those reported in Table 1. Also [Kornblith et al.: Do Better ImageNet Models Transfer Better? CVPR 2019: 2661-2671], which is not about improving transfer learning, reports results with their fine-tuning baseline (that would be your "None" in Table 1) that are better than the best results reported here among all the methods for ResNet 101, and this by a wide margin: Aircraft 10.4%, Flowers 7.5%, Pets 3.6%, DTD 2.6%, Caltech 8.3%.
> >
> > I understood that the purpose of Table 3 was to show the performance that would be achieved with your approach on a more SOTA transfer learning scheme, so it should be about genuine transfer. Reproducing past results on benchmarks that are now known to be flawed is not helpful in this regard.

---

> > > ### Author Response · Authors · 2020-11-24
> > > **Differences in experimental setup**
> > >
> > > Thanks for the comments. We agree there are lots of subtleties that make direct comparison across papers tricky, which is why we went to some effort to re-run all experiments in Tab 1 to make all methods exactly comparable.
> > >
> > > To recap the explanation for the differences:
> > >
> > > * First, please note that Kornblith numbers are NOT directly comparable to ours or DELTA/$\ell^2$-SP as Kornblith uses a different metric for the datasets mentioned with big discrepancies: namely, mean per-class, vs overall mean accuracy which we use in order to match with DELTA and $\ell^2$-SP.
> > > * Second, our main experiment in Tab 1 uses no data augmentation in order to focus on the impact of the regularizers proposed by each method. DELTA, Kornblith, and $\ell^2$-SP  use data augmentation, with the latter two using heavy augmentation of all kinds: color, geometry, etc.
> > > * Third, the Kornblith and DELTA papers use different and significantly stronger pre-trained weights. Kornblith's custom pre-trained ResNet-101 is significantly better than the publicly available variants, as explained in his Appendix A.3. We use the Keras pre-trained ResNet-101, which is based on the original ResNet-101. In contrast, DELTA uses the improved version of ResNet-101 included in torchvision. These architectural and training modifications present in the torchvision model are known to improve performance [1,2].
> > >
> > > We are happy to update the results to use data augmentation, etc, but we cannot do this now, given that this request was raised rather late in the discussion phase.
> > >
> > > [1] Goyal et al. (2017) Accurate, large  mini-batch SGD: training ImageNet  in 1 hour. arXiv.
> > >
> > > [2] Ross and Wilber (2016) Training and investigating residual nets. The Torch Blog.

---

### Decision · Program_Chairs · 2021-01-07
**Final Decision**

**Decision:**

Accept (Poster)

**Comment:**

This paper proposes constraints to be applied to the weights of a deep neural model during training. These constraints, motivated by an analysis of Rademacher complexity, are compared with other constraints and penalty approaches in transfer learning. The authors were able to build on the reviewers feedback to improve their paper on several points during the discussion phase, leading to a consensus for acceptance among reviewers. They also agreed to conduct experiments targeting stronger experimental results to compare all methods in the situation where they provide state-of-the-art results. This will make a useful contribution to the ICRL audience, and I recommend acceptance.